

# Prolonged pituitary down-regulation with full-dose of gonadotropin-releasing hormone agonist in different menstrual cycles: a retrospective cohort study

Yingfen Ying, Tanchu Yang, Huina Zhang, Chang Liu and Junzhao Zhao

Reproductive Medical Center, Department of OB & GYN, The Second Affiliated Hospital and Yuying Children's Hospital of Wenzhou Medical University, Wenzhou, Zhejiang, China

## ABSTRACT

**Background:** The efficiency of prolonged down-regulation caused by a full-dose of gonadotropin-releasing hormone agonist (GnRH-a) injected during different menstrual phases has not yet been researched. Our goal was to evaluate the effects of GnRH-a, which was used in different phases of the menstrual cycle in patients undergoing in vitro fertilization and embryo transfer.

**Methods:** This was a retrospective cohort study. A total of 320 patients received a prolonged pituitary down-regulated full-dose (3.75 mg) of triptorelin in the early follicular phase, and 160 patients received the same full-dose of triptorelin during the mid-luteal phase. Clinical and laboratory outcomes were compared between the two groups.

**Results:** The basic characteristics of the two groups were comparable. The mean number of retrieved oocytes, fertilized oocytes, cleavage oocytes and good quality embryos were comparable between the two groups. Although there was a higher antral follicle count, cyst formation rate, fertilization rate and cleavage rate in the follicular phase group, no statistically significant effects were seen on implantation rate (41.15% vs. 45.91%), clinical pregnancy rate (60.38% vs. 61.36%), ongoing pregnancy rate (57.74% vs. 57.58%), live birth rate (56.23% vs. 57.58%) or early abortion rate (2.64% vs. 3.79%) per fresh transfer cycle. Moreover, severe ovarian hyperstimulation syndrome rates at the early stage (1.89% vs. 2.27%) were low in both groups.

**Conclusions:** Prolonged pituitary down-regulation achieved by utilizing a full-dose of GnRH-a administrated in either phase of the menstrual cycle can have a positive effect on ongoing pregnancy rate and live-birth rate per fresh embryo transfer cycle. Ovarian cyst formation rate was higher in the follicular phase group, but this did not have any adverse impact on clinical results.

Corresponding author
Junzhao Zhao, joyce08@126.com

# INTRODUCTION

Over the last three decades, gonadotropin-releasing hormone agonists (GnRH-a) were the most commonly used drugs for controlled ovarian hyperstimulation (COH) in assisted
reproductive procedures. These drugs provided many benefits, such as the recovery of a larger number of oocytes, prevention of premature luteinizing hormone (LH) surge, luteinization, and a lower cycle cancellation rate (*Haydardedeoğlu & Kılıçdağ, 2016*; *Siristatidis et al., 2015*; *Xiao, Su & Zeng, 2014*). Utilizing GnRH-a has been considered the gold standard for COH. Different dosages and formulations of GnRH-a were devised. It has been noted that clinical pregnancy rate might increase when patients are given a period of artificial amenorrhea caused by GnRH-a (*Cai et al., 2018*; *Liao et al., 2015*; *Martínez et al., 2014*; *Ren et al., 2014*). One full-dose depot of long-acting GnRH-a per COH cycle would be more convenient for women than short-acting GnRH-a injections because there are fewer injections, leading to greater compliance as well as fewer incidences of infection (*Cheon et al., 2008*; *Gao et al., 2014*).

*Liao et al. (2015)* reported that clinical pregnancy rates in patients who received a depot GnRH-a regimen was higher than that of the daily low-dose group (57.5% vs. 46.9%). The prolonged regimen can be divided into two different types, according to the time that the GnRH agonist is administered. The first type, the follicular phase, is where the GnRH agonist is injected between the first and third days of the menstrual cycle; the second type, the mid-luteal phase, is where the drug is administered in the middle of the previous luteal cycle. It was concluded by *Broekmans et al., (1992)* that a single administration of GnRH-a either in the mid-luteal or the early follicular phase reached a high degree of similarity in pharmacodynamics response, which rapidly induced a reversible hypogonadotropic and hypogonadal condition. Pituitary and ovarian suppression was maintained until the beginning of the 8th week (*Broekmans et al., 1992*).

Until now, the clinical efficiency of prolonged down-regulation caused by a full-dose of GnRH-a injected during different menstrual phases has not yet been researched either at home or abroad. Does prolonged down-regulation caused by a full-dose of GnRH-a injected during different menstrual phases cause a similar high pregnancy rate? The aim of this study was to compare the clinical efficacy of a full-dose of the GnRH-a drug triptorelin in follicular and mid-luteal regimens before COH through ongoing pregnancy and live birth, which are vital indicators in suitable infertile couples. We can then explore whether one or both meet the clinical requirements, and which one is better.

## MATERIALS AND METHODS

### Study participants

This was a retrospective study approved by the ethics committee of the Second Affiliated Hospital of Wenzhou Medical University (Ethic Reference No: L-2018-19). This study was performed on fresh IVF/ICSI ET cycles with 3.75 mg triptorelin for prolonged pituitary down-regulation injected in the follicular phase (Group 1) or mid-luteal phase (Group 2) of the menstrual cycle from June 2016 to May 2017. Inclusion criteria were: (1) all patients were 20–38 years old with infertility due to: salpingemphraxis, endometriosis, polycystic ovarian syndrome, polycystic ovary, male factors, or idiopathic causes; (2) a body mass index (BMI) <28 kg/m$^2$; (3) basal serum follicle stimulating hormone (FSH) <12 IU/L and estradiol (E$_2$) <80 pg/mL were determined on the third day of the cycle. Uterine abnormalities such as müllerian malformations, fibroids and

adenomyosis were excluded. Patients that had received any ovulation-induction treatment within 3 months of study entry were also excluded. Different treatment regimens were performed simultaneously for the two groups of patients. Before a cycle was initiated, candidate patients were fully informed with detailed information of both protocols, including the time-point and duration of down-regulation, the pregnancy rate, and the potential risk of strong pituitary depression. Based on this information, patients made the decision to get prolonged pituitary down-regulation in the early follicular phase or in the mid-luteal phase. All the patients enrolled gave written informed consent for the whole procedure.

Of the 480 cycles that met the study criteria, 25 cycles in Group 1 and seven cycles in Group 2 were cancelled because fewer than one oocyte was retrieved, and subsequently there were no available embryos. An additional 30 cycles in Group 1 and 21 cycles in Group 2 were canceled because a freeze-all of the embryos was performed. Therefore, 265 cycles of Group 1 and 132 cycles of Group 2 were observed (Fig. 1).

## COH and IVF/ICSI procedure

Pituitary desensitization was achieved with a single full-dose injection of 3.75 mg (Triptorelin; Ferring, Kiel, Germany) during the follicular phase of the menstrual cycle in Group 1, in which ovarian stimulation would occur 32–38 days later. In Group 2, pituitary down-regulation was started in the mid-luteal phase with a full dose of 3.75 mg triptorelin, and COH with gonadotropin would commence from 32 to 38 days after the single GnRH-a injection.

Successful pituitary down-regulation was confirmed after finding no antral follicles larger than eight mm, $E_2$ concentration less than 50 pg/mL, serum LH levels less than five IU/L, and endometrium thickness less than five mm. COH with recombinant FSH (Gonal-F; Merck Serono, Aubonne, Switzerland) was started at one to four ampules (75–300 IU) and the dosage was adjusted according to patients' BMI, antral follicular count (AFC), basal FSH level and follicular growth response. Transvaginal ultrasound and serum $E_2$, LH and P levels were used to monitor ovarian response. Recombinant LH (r-LH) (recombinant-LH; Merck Serono, Aubonne, Switzerland) was added at the late stage of follicular growth when the LH level was <0.5 IU/L with the dosage of 75 IU r-LH. One dose of 5,000–10,000 IU human chorionic gonadotropin (hCG; Livzon, Guangdong, China) was given when at least two follicles reached 18 mm in mean diameter under ultrasonograph. Oocytes were retrieved 34–36 h later via vaginal ultrasound.

Oocytes were fertilized by IVF or ICSI 4–6 h after oocyte retrieval. Embryos were incubated at 37 °C under humidified gas phase of a mixture of 6% $CO_2$, 4% $O_2$ and 90% $N_2$. Embryo score was performed as described (Alpha Scientists in Reproductive Medicine and ESHRE Special Interest Group of Embryology, 2011). Normal fertilization was confirmed by the presence of two pronuclear and two polar bodies 16–18 h post-insemination. Cleavage-stage embryo score was based on the number of blastomeres, blastomere size and the proportion of fragments. Morulae-stage embryos were assessed on the basis of the proportion of compaction. Blastocyst scoring was consisted of the expansion stage of blastocyst cavity expansion, density, cell number of inner cell mass

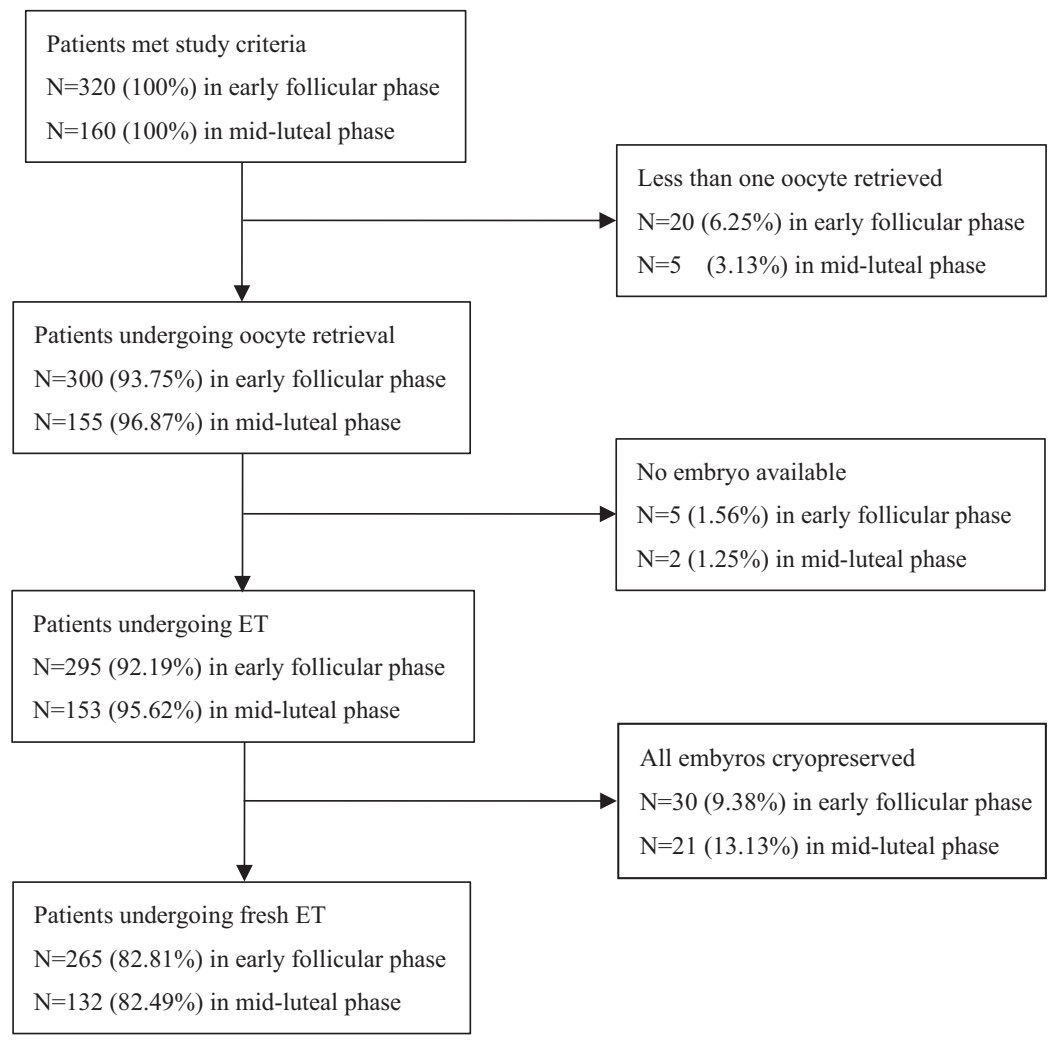

**Figure 1 Flow chart describing drop out of enrolled patients.** Of the 480 cycles eligible for the study, 20 cycles in Group 1 and five cycles in Group 2 were canceled because fewer than one oocyte was obtained; five cycles in Group 1 and two cycles in Group 2 were cancelled because of no available embryos. Finally, another 30 cycles in Group 1 and 21 cycles in Group 2 were canceled due to the freezing all of the embryos. So 265 cycles in Group 1 and 132 cycles in Group 2 were observed.

and the cohesion as well as regularity of trophectoderm (*Alpha Scientists in Reproductive Medicine and ESHRE Special Interest Group of Embryology, 2011*). Embryo transfer was determined according to the embryo quality, the state of endometrium and the physician's convenience. Under the guidance of abdominal ultrasound, embryo transfer was performed with at least one good quality embryo at the specific developmental stage. Luteal phase support was sustained with micronized progesterone (Utrogestan; Capsugel, Besins Manufacturing Belgium, Bruxelles, Belgium), 200 mg orally administrated per day, and progesterone vaginal gel prolonged release (Crinone; Merck Serono, Hertfordshire, United Kingdom), 90 mg each day from the day of oocyte retrieval. Procedures of Group 1 and Group 2 were shown in Figs. 2 and 3 respectively. Serum β-hCG determination was performed 14 days after cleavage-stage embryo transfer, or

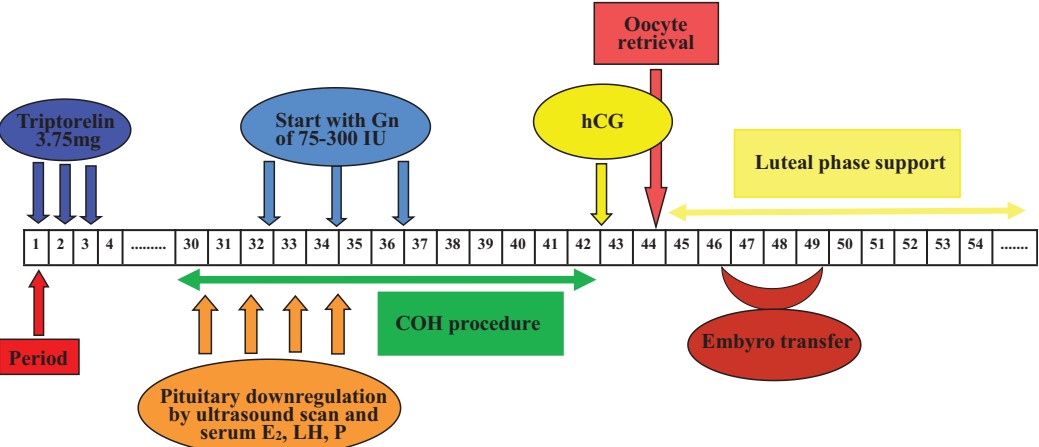

**Figure 2** **Flow chart of GnRH agonist prolonged regimen in Group 1.** One depot of 3.75 mg Triptorelin was injected from 1st to 3rd days of the menstrual cycle, in which ovarian stimulation with 75–300 IU Gns would start 32–38 days later along with confirmation of pituitary downregulation. COH procedure was adjusted according to patients' ovarian response. One dose of hCG was given when at least two follicles reached 18 mm in mean diameter, and then oocytes were retrieved 34–36 h later. After that, embryo transfer was performed 2–5 days later. Luteal phase support was sustained from the day of oocyte retrieval. (COH, controlled ovarian hyperstimulation; $E_2$, estradiol; Gns, gonadotropins; hCG, human chorionic gonadotrophin; LH, luteinizing hormone; P, progesterone).

12 days after blastocyst transfer. Clinical pregnancy was defined as a transvaginal ultrasound detection of gestational sac(s) with pulsating heart beats 4–5 weeks after embryo transfer. Implantation rate was calculated as the ratio of the number of gestational sac(s) over the number of transferred fresh embryos. Early abortion was defined as a clinical pregnancy failing to reach the 12th gestational week. Ongoing pregnancy was defined as a clinical pregnancy that reached more than 12 gestational weeks.

### Statistical analysis

The demographic data of the infertile patients, serum hormonal levels, stimulation requirements and clinical outcome variables were all compared between the two groups. Analysis was performed with SPSS (version 22.0; IBM, Armonk, NY, USA). Independent samples $t$-test or Mann–Whitney test were used for continuous variables. Dichotomous variables were analyzed by Chi-square test or Fisher's exact test as required. Multivariable logistic regression analysis was used to test the association between the selected variables and the probability of live-birth rates. In all analysis, $P < 0.05$ was considered statistically significant.

### RESULTS

The general characteristics of the patients of both groups were comparable. The average age (30.58 ± 3.57 vs. 31.14 ± 4.18 yrs) and the mean duration of infertility (3.58 ± 2.44 vs. 3.80 ± 2.62 yrs) were similar in both groups. There were no significant differences in the BMI (BMI: 21.28 ± 2.56 vs. 21.79 ± 2.65) nor the proportion of primary infertility between the two groups. The basal FSH level of Group 1 was less than that of Group 2,

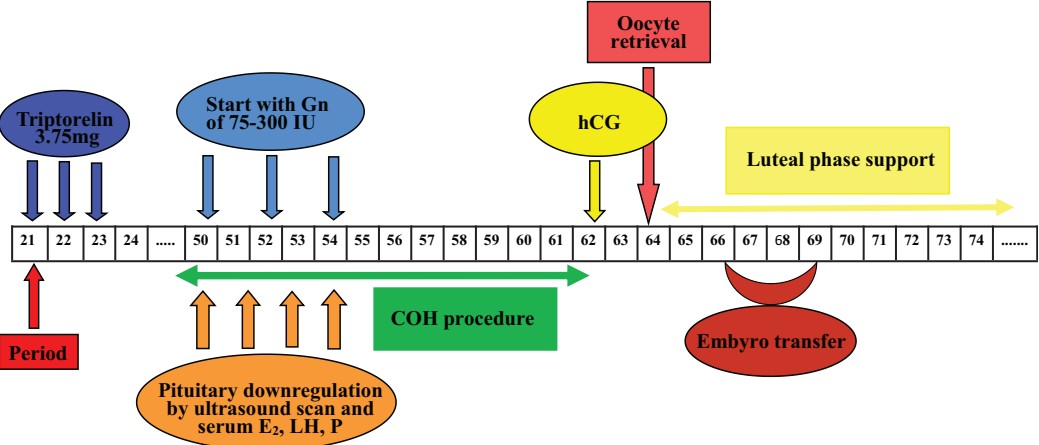

**Figure 3 Flow chart of GnRH agonist prolonged regimen in Group 2.** One depot of 3.75 mg Triptorelin was injected from mid-luteal phase of last menstrual cycle, the rest procedures were similar to those of Group 1. COH, controlled ovarian hyperstimulation; $E_2$, estradiol; Gns, gonadotropins; hCG, human chorionic gonadotrophin; LH, luteinizing hormone; P, progesterone.

though without significant difference between the patient cohorts (Table 1). There was more AFC on the day COH commenced in Group 1 (14.68 ± 5.57 vs. 12.61 ± 5.64, $P = 0.001$; Table 1). Considering the basal FSH level and AFC, the ovarian reserve function of Group 1 was better than that of Group 2. The average serum LH, $E_2$ and P levels were similar in both groups 32–38 days after pituitary desensitization. The serum LH, P, $E_2$ levels and endometrial thickness were equivalent on the day of hCG in both groups. The duration of ovarian simulation and the total dosage of ovarian stimulating drug were similar in both treatment protocols (Table 1).

The number of oocytes retrieved, fertilized, and cleaved, and the number of good quality embryos in both procedures were similar. However, the ratio of fertilization and cleavage was significantly different between treatment groups (68.81% (2,116/3,075) vs. 65.67% (1,058/1,611) $P = 0.03$), (97.26% (2,058/2,116) vs. 94.61% (1,001/1,058) $P < 0.001$) (Table 2). The rate of ovarian cyst formation was higher in Group 1 (6.42% (17/265) vs. 0% (0/132) $P = 0.003$) (Table 2).

Clinical pregnancy, embryo transfer, ongoing pregnancy, early abortion and live birth rates were comparable between treatment protocols. The incidence of severe ovarian hyperstimulation syndrome (OHSS) at an early stage was not significantly different between groups. The proportion of freezing all embryos was equivalent in both groups, 10.17% vs. 13.73% (Table 2).

We divided Group 1 into two subgroups, with and without cyst development, and compared their laboratory and clinical outcomes. The proportion of good quality embryos, clinical pregnancy rate, ongoing pregnancy rates, early abortion rate and live birth rate were all higher in the cyst subgroup, but the differences were not statistically significant (Table 3). The average $E_2$ level on the starting day in the cyst subgroup (28.71 ± 11.20 pg/mL) was more comparable than that in the non-cyst subgroup. Only one patient's $E_2$ level on the starting day was more than 50 pg/mL not shown in

**Table 1 Patients' basic characteristics and stimulation variables of the two groups.**

|  | Group 1 ($n = 265$) | Group 2 ($n = 132$) | P-value |
|---|---|---|---|
| **Patient characteristics** |  |  |  |
| Classifications of infertility |  |  | 0.915 |
|    Primary infertility %(n) | 44.15 (117/265) | 43.18 (57/132) | 0.855 |
|    Secondary infertility %(n) | 55.85 (148/265) | 56.82 (75/132) | 0.855 |
|    Age (years) | 30.58 ± 3.57 | 31.14 ± 4.1858 | 0.134 |
|    BMI (Kg/m$^2$) | 21.28 ± 2.56 | 21.79 ± 2.65 | 0.063 |
|    Infertility durations (years) | 3.58 ± 2.44 | 3.80 ± 2.62 | 0.998 |
| **IVF testifies** |  |  |  |
| Female infertility %(n) | 48.68 (129/265) | 43.94 (58/132) | 0.373 |
|    Polycystic ovary syndrome %(n) | 8.30 (22/265) | 3.03 (4/132) | 0.052 |
|    Endometriosis %(n) | 10.18 (27/265) | 4.55 (6/132) | 0.056 |
| Male infertility %(n) | 16.98 (45/265) | 14.39 (19/132) | 0.509 |
| Couple reasons %(n) | 28.67 (76/265) | 39.39 (42/132) | 0.519 |
| Unkown reasons %(n) | 5.67 (15/265) | 2.28 (3/132) | 0.126 |
| **Basic hormones on the day 3** |  |  |  |
| D3 LH (IU/L) | 4.24 ± 2.48 | 4.18 ± 1.92 | 0.373 |
| D3 FSH (IU/L) | 5.90 ± 1.68 | 6.22 ± 1.78 | 0.085 |
| D3 E$_2$ (pg/mL) | 44.56 ± 14.12 | 45.80 ± 16.01 | 0.432 |
| **On the commencing day** |  |  |  |
| LH level (IU/L) | 0.49 ± 0.24 | 0.46 ± 0.33 | 0.082 |
| E$_2$ level (pg/mL) | 29.26 ± 12.19 | 26.90 ± 9.91 | 0.121 |
| P$_4$ level (ng/mL) | 0.35 ± 0.17 | 0.42 ± 0.22 | 0.373 |
| Antral follicular count (n) | 14.68 ± 5.57 | 12.61 ± 5.64 | 0.001 |
| **On the day of hCG** |  |  |  |
| LH level (IU/L) | 0.63 ± 0.43 | 0.60 ± 0.54 | 0.583 |
| E$_2$ level (pg/mL) | 1,994.03 ± 1,220.48 | 2,046.86 ± 1,128.08 | 0.756 |
| P$_4$ level (ng/mL) | 0.54 ± 0.32 | 0.77 ± 1.84 | 0.536 |
| Endometrial thickness (mm) | 11.37 ± 2.03 | 11.32 ± 2.26 | 0.264 |
| Initial dose of FSH (IU/day) | 166.09 ± 53.81 | 168.56 ± 50.01 | 0.530 |
| Days of stimulation (days) | 11.51 ± 3.55 | 12.41 ± 3.01 | 0.282 |
| Total dose of FSH (IU) | 2,243.97 ± 1,070.13 | 2,440.08 ± 1,039.48 | 0.156 |

Table 3. Initial and total doses of FSH were higher in the cyst subgroup, but there was no statistical difference (Table 3).

To evaluate the contribution of different time-points of down-regulation on the difference in live-birth rate, a multivariate logistic regression model was used to analyze some related confounding factors. The variables were basal FSH levels, AFC, P, LH and E$_2$ levels on the commencing day of stimulation, and endometrial thickness on the day of hCG. Moreover, other factors (age, BMI, progesterone elevation on the hCG day, number of good quality embryos on day 3) were also selected based on clinical experiences, though they were similar in both cohorts. The Hosmer–Lemshow goodness-of-fit Chi-square test statistic was 4.099 ($P = 0.848$), which suggested that the multivariable

**Table 2 Laboratory and clinical outcomes of the two groups.**

|  | Group 1 ($n = 265$) | Group 2 ($n = 132$) | P-value |
|---|---|---|---|
| **Laboratory outcomes** | | | |
| IVF %(n) | 64.15 (170/265) | 70.45 (93/132) | 0.211 |
| ICSI %(n) | 27.17 (72/265) | 18.18 (24/132) | 0.062 |
| IVF/ICSI %(n) | 8.68 (23/265) | 11.36 (15/132) | 0.392 |
| No. of oocytes retrieved | 11.60 ± 5.49 | 12.20 ± 4.91 | 0.288 |
| No. of 2PNs per cycle | 7.98 ± 4.31 | 8.02 ± 3.87 | 0.946 |
| No. of cleavage per cycle | 7.77 ± 4.28 | 7.58 ± 3.78 | 0.678 |
| No. of good quality embryos per cycle | 4.19 ± 3.20 | 4.08 ± 3.05 | 0.745 |
| Rate of fertilized eggs %(n) | 68.81 (2,116/3,075) | 65.67 (1,058/1,611) | 0.030 |
| Rate of cleavage %(n) | 97.26 (2,058/2,116) | 94.61 (1,001/1,058) | <0.001 |
| Rate of good quality D3 embyros %(n) | 50.73 (1,111/2,190) | 50.99 (539/1,057) | 0.911 |
| **Clinical outcomes** | | | |
| No. Embryos transfer | 2.03 ± 0.34 | 1.95 ± 0.31 | 0.904 |
|    D2–D3 % (mean days) | 73.20 (2.97 ± 0.16) | 63.42 (2.96 ± 0.21) | 0.091 |
|    D4–D5 % (mean days) | 26.80 (4.35 ± 0.48) | 36.58 (4.40 ± 0.50) | 0.383 |
| Clinical pregnancy rate %(n) | 60.38 (160/265) | 61.36 (81/132) | 0.913 |
| Ongoing pregnancy rate %(n) | 57.74 (153/265) | 57.58 (76/132) | 1.000 |
| Live birth rate %(n) | 56.23 (149/265) | 57.58 (76/132) | 0.830 |
| Early abortion rate %(n) | 2.64 (7/265) | 3.79 (5/132) | 0.543 |
| Implantation rate %(n) | 41.15 (221/537) | 45.91 (118/257) | 0.220 |
| Ovarian cyst formation rate %(n) | 6.42 (17/265) | 0 (0/132) | 0.003 |
| Rate of severe early OHSS %(n) | 1.89 (5/265) | 2.27 (3/132) | 0.797 |
| Rate of $E_2$ > 4,000 (pg/mL) on the day of hCG %(n) | 6.04 (16/265) | 7.56 (10/132) | 0.560 |
| Rate of all embyro cryopreserved %(n) | 10.17 (30/295) | 13.73 (21/153) | 0.664 |

model was a good fit ($P > 0.05$). Adjusted by the previously mentioned potential confounding factors, the odds ratio of prolonged down-regulation in the early follicular phase vs. mid-luteal phase on the live-birth rate was 1.040 (95% confidence interval [0.997–1.084]).

# DISCUSSION

As expected and suggested by some researchers, a long-acting protocol with a full-dose of GnRH-a caused pituitary side effects, inhibited ovarian steroidogenesis, and affected the differentiation of granulosa cells (*Dor et al., 2000*; *Metallinou et al., 2012*). One review (*Albuquerque et al., 2013*) indicated that using a long-acting GnRH-a for pituitary down-regulation required more gonadotropins and a higher duration of ovarian stimulation. However, our data showed that the total dosage of gonadotropins in our study was lower in Group 1, 2,243 IU vs. 2,440 IU in Group 2, both of which were slightly lower or similar to that of one-third or one-half dose of the GnRH-a depot published by certain studies (*Chen et al., 2016*). That means a full-dose of GnRH agonist would not drastically increase the economic burden for patients. In addition, the effective duration of

**Table 3 Data comparison between cyst and non-cystic subgroups in Group 1.**

|  | Cyst subgroup (n = 17) | Non-cyst subgroup (n = 248) | P-value |
|---|---|---|---|
| **Basic situation** | | | |
| Age (years) | 31.39 ± 2.98 | 30.53 ± 3.61 | 0.333 |
| BMI (Kg/m$^2$) | 20.92 ± 2.24 | 21.31 ± 2.59 | 0.351 |
| Infertility durations (years) | 4.18 ± 3.34 | 3.54 ± 2.37 | 0.303 |
| **Basic hormones on cycle day 3** | | | |
| D3 LH (IU/L) | 5.17 ± 2.97 | 4.18 ± 2.44 | 0.109 |
| D3 FSH (IU/L) | 6.28 ± 1.77 | 5.87 ± 1.68 | 0.336 |
| D3 $E_2$ (pg/mL) | 49.58 ± 15.67 | 44.22 ± 13.95 | 0.130 |
| **On the commencing day** | | | |
| LH level (IU/L) | 0.54 ± 0.22 | 0.49 ± 0.25 | 0.379 |
| $E_2$ level (pg/mL) | 28.71 ± 11.20 | 29.30 ± 12.28 | 0.847 |
| $P_4$ level (ng/mL) | 0.33 ± 0.15 | 0.36 ± 0.18 | 0.624 |
| Antral follicular count (n) | 12.88 ± 5.49 | 15.01 ± 5.56 | 0.128 |
| **On the day of hCG** | | | |
| LH level (IU/L) | 0.54 ± 0.27 | 0.64 ± 0.44 | 0.343 |
| $E_2$ level (pg/mL) | 1,609.00 ± 1,276.31 | 2,020.42 ± 1,214.77 | 0.179 |
| $P_4$ level (ng/mL) | 0.58 ± 0.41 | 0.54 ± 0.31 | 0.555 |
| Endometrial thickness (mm) | 11.45 ± 2.49 | 11.37 ± 2.00 | 0.875 |
| Initial dose of FSH (IU/day) | 183.82 ± 54.44 | 164.93 ± 53.59 | 0.161 |
| Days of stimulation (days) | 11.88 ± 1.54 | 11.48 ± 3.65 | 0.652 |
| Total dose of FSH (IU) | 2,585.29 ± 950.25 | 2,220.57 ± 1,075.61 | 0.174 |
| **Laboratory data** | | | |
| No. of oocytes retrieved | 11.00 ± 4.89 | 11.65 ± 5.53 | 0.640 |
| No. of 2PNs per cycle | 7.41 ± 4.42 | 8.02 ± 4.31 | 0.572 |
| No. of cleavage per cycle | 7.35 ± 4.33 | 7.78 ± 4.28 | 0.682 |
| No. of good quality embryos per cycle | 4.41 ± 3.34 | 4.18 ± 3.20 | 0.771 |
| Rate of good quality embyros %(n) | 53.96 (75/139) | 50.51 (1,036/2,051) | 0.483 |
| **Clinical outcomes** | | | |
| Clinical pregnancy rate %(n) | 82.35 (14/17) | 58.87 (146/248) | 0.072 |
| Ongoing pregnancy rate %(n) | 76.47 (13/17) | 56.45 (140/248) | 0.131 |
| Live birth rate %(n) | 76.47 (13/17) | 54.84 (136/248) | 0.082 |
| Early abortion rate %(n) | 5.88 (1/17) | 2.42 (6/248) | 0.375 |
| Rate of E2 > 4,000 (pg/mL) on the day of hCG %(n) | 5.88 (1/17) | 6.05 (15/248) | 0.978 |

gonadotropins with respect to ovarian stimulation was 11.51 days for Group 1 vs. 12.41 days for Group 2, which was similar to the duration of half-dose of GnRH agonist in long-acting regimens (*Ren et al., 2013*). Moreover, the clinical and ongoing pregnancy rates achieved in this study were higher than those of procedures involving one-half dose or one-third dose of GnRH-a (*Chen et al., 2016*).

Optimal serum LH concentrations at the beginning of ovarian stimulation after pituitary down-regulation with GnRH agonist is neither <0.1 IU/L nor >1 IU/L, which are

the levels of LH required in follicular development (*Shahrokh Tehraninejad et al., 2017*; *Mochtar et al., 2017*). Excessive pituitary suppression, meaning less than 0.1 IU/L serum LH levels, requires exogenous LH to increase the number of mature oocytes and good quality embryos (*Razi et al., 2014*; *Shahrokh Tehraninejad et al., 2017*). In contrast, insufficient pituitary suppression, meaning more than one IU/L of serum LH concentration, has been proven to have adverse impacts on oocyte and embryo quality and reduces embryo implantation potential and clinical pregnancy rates (*Loumaye et al., 1989*). Our data showed that serum LH levels at 32–38 days of pituitary suppression were 0.46–0.49 IU/L, appropriate LH concentrations for starting COH.

Our data simultaneously showed that serum LH concentrations on the day of hCG injection were low in patients in both procedures, which was similar to the study of *Mao et al. (2014)*, indicating positive endometrial receptivity. However, little is known about the effects of LH on the endometrium regarding activation in IVF/ICSI cycles (*Londra et al., 2016*). The corresponding risk of severe early OHSS after using a long-acting GnRH-a protocol was 1–6%, similarly found in our research (*Albuquerque et al., 2013*).

One of the detrimental effects of using GnRH-a is the formation of ovarian cysts, which may be either functional or non-functional. In this research, prolonged pituitary down-regulation throughout the early follicular phase was shown to create many more ovarian cysts, most of which were non-functional. The pathophysiological mechanism of cyst formation is unknown so far. However, possible mechanisms have been published (*Feldberg et al., 1989*; *Pereira et al., 2016*; *McDonnell, Marjoribanks & Hart, 2014*). It is easy for ovarian cysts to form when initiated at the follicular phase, especially in women with higher basal FSH levels during their menstrual cycle. Some other explanations could be as follows: the effect of primary flare-up caused by the effect of GnRH-a on gonadotropins; inadequate suppression of circulated gonadotropins following hypophyzectomy; and the direct effect of GnRH agonists on ovaries and subsequent steroidogenesis (*Firouzabadi, Sekhavat & Javedani, 2010*). Similarly, in this research, ovarian cyst formation rate was higher in the group in which GnRH-a was administrated in the early follicular phase. While some studies proposed that ovarian cyst formation not only compromised the quality of oocytes and embryos but also increased cycle cancelation rates and decreased pregnancy rates, these results were contrary to our findings (*Qublan et al., 2006*).

With one exception, all cysts were nonfunctional. We did not treat non-functional cysts, although treatments such as draining the cyst via ultrasound-guided cyst puncture are available. Though the initial dose and total dose of samples were much higher for the non-cyst subgroup, the rates of good quality embryos, clinical pregnancy rate, continued pregnancy rates and live birth rate were all slightly but not significantly higher in the cyst subgroup. These results were consistent with some studies in the literature (*Firouzabadi, Sekhavat & Javedani, 2010*; *Qublan et al., 2006*). Most participants in a survey of worldwide IVF centers believe that the presence of non-functional ovarian cysts does not influence the outcome and will therefore start stimulation, while 28% of centers will go forward with the cycle only after aspirating this cyst (*Tur-Kaspa & Fauser, 2013*).

Therefore, we believe that the non-functional ovarian cysts formed after GnRH agonist do not affect the clinical outcomes. The effect of functional ovarian cysts on clinical outcomes after GnRH-a usage is controversial. There is insufficient evidence to determine whether drainage of functional ovarian cysts prior to COH negatively influenced live birth rate, clinical pregnancy rate, number of follicles recruited or oocytes collected in women with a functional ovarian cyst (*McDonnell, Marjoribanks & Hart, 2014*). In this research, the formation rate of functional ovarian cysts was much lower than that of nonfunctional cysts.

## CONCLUSION

In conclusion, our data suggested that prolonged pituitary down-regulation in the early follicular phase can be as equally effective as that in the mid-luteal phase in fresh IVF/ICSI-ET cycles. Ovarian cyst formation rate was higher in the follicular phase group, while this group still showed comparable efficiency in the results of IVF performance and clinical outcomes. Additional well-designed randomized trials that compare long-acting GnRH agonists injected in the follicular phase and in the mid-luteal phase are expected to demonstrate good IVF/ICSI outcomes.

### Funding

This work was supported by the Key Scientific and Technological Innovation Team Foundation of Wenzhou, Zhejiang, China (No. C20170007). The funders had no role in study design, data collection and analysis, decision to publish, or preparation of the manuscript.

### Grant Disclosures

The following grant information was disclosed by the authors:
Key Scientific and Technological Innovation Team Foundation of Wenzhou, Zhejiang, China: C20170007.

### Competing Interests

The authors declare that they have no competing interests.

### Author Contributions

- Yingfen Ying performed the experiments, analyzed the data, contributed reagents/materials/analysis tools, prepared figures and/or tables, authored or reviewed drafts of the paper, approved the final draft.
- Tanchu Yang analyzed the data, approved the final draft.
- Huina Zhang contributed reagents/materials/analysis tools, approved the final draft.
- Chang Liu prepared figures and/or tables, approved the final draft.
- Junzhao Zhao conceived and designed the experiments, approved the final draft.

## Human Ethics

The following information was supplied relating to ethical approvals (i.e., approving body and any reference numbers):

This retrospective study was approved by the ethics committee of the Second Affiliated Hospital of Wenzhou Medical University (L-2018–19).

## Data Availability

Raw data is available in the Supplemental Files.

## Supplemental Information

Supplemental information for this article can be found online at http://dx.doi.org/10.7717/peerj.6837#supplemental-information.

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
