# Peer review of "Prolonged pituitary down-regulation with full-dose of gonadotropin-releasing hormone agonist in different menstrual cycles: a retrospective cohort study"

_PeerJ, doi:10.7717/peerj.6837_

## Round 0.1 · original submission · Major Revisions

Dear Authors,
The Reviewers found your manuscript very interesting, howover they recommend a thorough revision in order to achieve publication.
I would suggest to take into consideration the Reviewers' comments, discuss and incorporate them within your manuscript in order to reach the standard requested for publication.
Best regards

Salvatore Andrea Mastrolia
PeerJ Academic Editor

Reviewer 1 ·

Basic reporting

no comment

Experimental design

1. The manner of patient assignment is unknwon. Because a retrospective study may be skewed by various confoundering factors and selection biases, knowning how the patietns were assigned to different treatment is important to understand the study.
2. Patients with PCOS and/or endometriosis were included. Because these conditions were well known to affect the IVF outcomes, the proportions of patients with PCOS and endometriosis should be given in Table 1.
3. The authors may mentioned the dosage and timing of the use of recombinant LH.
4. The authors mentioned that embryos were transferred 3-6 days after OPU. It means that patietns would recive either blastocyst transfer or cleavage stage transfer. For comparting the IVF outcomes following a single transfer, knowing the number of embryos transferred, the stage of embryos transferred and the quality of embryos transferred is important.

Validity of the findings

1. The study used only univariate analyses when the patients` characteristics and treatment outcomes were compared between different groups. Some of the results seems to be marginal. For instance, the P value for day 3 FSH was 0.085. It may imply that the significance would reach the criteria if the sample size was enlarged. It was also supported by the a difference in AFC (P<0.001). Taken together, it may suggest that patients in Group2 have a lower ovarian reserve thatn those in Group1. Considering the potential difference between Groups, mulivairate analyses should be taken.
2. Other factors that related to the IVF outcomes should be considered. The outcomes should be adjusted for the factors such as the number of embryos transferred.
3. The author may reorganized the iterm "IVF testifies" to yiel meaningful information. For instance, the term "femal infertility" may include tubal infertility or PCOS patients, but infertile patients with different etiologies may have different prognosis with regard to IVF outcomes.

Additional comments

In their retrospective cohort study, the authors compared the cinical and laboratory outcomes beween 320 patients received a prolonged pituitary down-regulated full-dose (3.75mg) of triptorelin in the early follicular phase and 160 patients received the same full-dose of triptorelin during the mid-luteal phase.
Although the outcomes of long luteal and long follicular phase GnRH agonist protocols have been investigated, less is knwon in patients reciveing prolonged prolonged pituitary down-regulation with full-dose GnRH agonist. The study highlighted the difference in cyst formation rate, but was limited by its sample size and retrospective nature.

·

Basic reporting

This study retrospectively investigated the clinical pregnancy outcome of a total of 480 subjects who underwent long GnRH-a downregulation at different time for IVF/ICSI treatment. The study design was clear and the data presented were reasonable. However most of the references were out of date, with only 2 papers (out of 32 papers in total) were published within 3 recent years. I recommend the authors should update their references as so many new papers published each year in reproductive area.

Experimental design

This was a retrospective study, however, the authors did not clarify the study period clearly in the materials part.

Validity of the findings

The findings that the pregnancy outcomes had no significant difference between women starting long GnRHa downregulation since early follicular phase and women starting long GnRHa downregulation since mid-luteal phase, did provide a certain useful information for clinical doctors and researchers.

Additional comments

no comment

Reviewer 3 ·

Basic reporting

Need to further justify why any clinic would choose to use a long acting GnRHa in late lutela phase. The rationale for conducting the study is not justified adequately.

Experimental design

Upon going through the study design and the process of conduct including the ethics committee approval and prospectively taking informed consent, this appears to be a prospective study however the title mentions it as retrospective analysis.

Figure 3 which describes the stimulation protocol with group 2 started on GnRHa in mid lutes phase shows that patients had periods. Either it’s a misrepresentation or needs explanation.

Details on the number of embryos transferred not mentioned and also feta love reduction if any.

Validity of the findings

Does the data include subjects who experienced OHSS? Was that early or late?
I do not see them accounted for in the total sample size and drop outs/exclusion.

Additional comments

no description of the number of embryos transferred

---

## Round 0.2 · Major Revisions

Dear Authors,

The Reviewers found your manuscript very interesting, moreover there are still major concerns highlighted by one of the Reviewers.

I would suggest to take into consideration the Reviewers' comments, discuss and incorporate them within your manuscript in order to reach the standard requested for publication.

Best regards

Salvatore Andrea Mastrolia
PeerJ Academic Editor

Reviewer 1 ·

Basic reporting

The basic reporting has been improved by adding detailed information.

Experimental design

The methology has been improved following revision.

Validity of the findings

In the revison, the authors provided further information to support their conclusion.

Additional comments

The manuscript has been revised according to Reviewer`s input.

Reviewer 3 ·

Basic reporting

The changes are well incorporated. No more comments

Experimental design

No more comments

Validity of the findings

No more comments

Additional comments

No more comments

Reviewer 4 ·

Basic reporting

This manuscript did a retrospective controlled study to investigate the efficiency of an injection of GnRH agonist in the early follicular phase versus that in the mid-luteal phase for ART cycles. The topic is interesting for clinical practice. However, most references focused on patients with endometriosis, not for general infertile couples. The authors should make this point clearly in the manuscript.

Experimental design

This is a retrospective controlled study to compare a full dose of GnRH agonist injection at early follicular phase with that at mid-luteal phase. Since the basic characteristics (antral follicle count) between the two groups. The choice or assignment of the two protocol might be clarified in the section of material and method.

Validity of the findings

The statistics regarding comparison between the two groups is sound. However, the conclusion is beyond the data analysis. The analysis only showed a full-dose of GnRH agonist injection at the early follicular phase is as effective as that in mid-luteal phase. I did not see any data about these two protocols are better than other protocols in terms of ongoing pregann. The authors may have to clarify the startment,

Additional comments

This manuscript did a retrospective controlled study to investigate the efficiency of an injection of GnRH agonist in the early follicular phase versus that in the mid-luteal phase for ART cycles.

1. In the abstract, this was a retrospective cohort study. However, the title described a retrospective controlled study. Please clarify it and keep in consistency.
2. In the abstract, authors declared that prolonged pituitary down regulation in early follicular or mid-luteal phase can “improve” ongoing pregnancy rate and live-birth rate. The data and design of this study couldn’t support the conclusion. Is this the authors’ speculation?

Introduction:
1. Line 61: The benefit of GnRH agonist for IVF cycles exhibited mainly for patients with endometriosis. Please state clearly in the related references (Surrey, 2015; Tanbo & Fedorcsak, 2017). The reference (Surrey, 2002) is also focused on patients with endometriosis. The studied population in this manuscript is not limited in patients with endometriosis. The references here might not justify for the use of prolonged pituitary down regulation for most infertile couples.

Materials and methods:
1. How to assign the patients into group 1 or group 2? Who will receive early follicular phase injection? Who will receive mid-luteal phase injection? The antral follicle count (ovarian reserve) is significantly higher in group 1 than group 2.
2. Line 135: How much is the dose of micronized progesterone per day? 200mg per day or more?

Results:
1. Please add the percentage data into the flow chart (Figure 1) to make it more clearly and easily for reading.
2. The period in Figure 3 is strange. Do you mean that the period is on day 21 or on the 21 st day from period?

Discussion:
1. Line 203: Again, all the references (Tanbo & Fedorcsak, 2017; Surrey et al., 2002; Soritsa et al., 2015) here are works on patients with endometriosis. The population in this manuscript is more general infertile couples, which may be not comparable with the patients with endometriosis. In addition, the first paragraph in the section of Discussion may be more suitable in the section of Introduction.
2. Line 252: In general, the oocyte or granulosa cells in the primordial follicles have no FSH or LH receptors. How can the primordial follicle be responsive to GnRH-a or gonadotropin stimulation? Please explain or give a reference for this point.

Conclusion
1. Line 277-278: The authors said, “In conclusion, our data suggested that prolonged pituitary down-regulation in the early follicular phase or in mid-luteal phase of the cycle can achieve higher rates of ongoing pregnancy and live birth, with low early abortion rates in fresh IVF/ICSI-ET cycles.” What does that statement “..higher rates of ongoing pregnancy….” mean? Higher than what protocol? The data in this manuscript did not support such conclusion.

---

## Round 0.3 · Major Revisions

Dear Authors,

After the previous editorial decision, and the response provided by the corresponding author of this manuscript, the case has been thoroughly discussed within the editorial board, due to appreciation that we have for all our potential contributors and the relevance we give to their opinion.

We respectfully disagree with the corresponding author regarding the fact that, in order to get accepted for publication, an indefinite number of revisions should be allowed on a manuscript unless or until reviewers clearly state that it must be rejected. The decision to publish a manuscript is the result of a number of conditions that needs to be fulfilled, and some manuscripts do not even reach the peer review phase if, after editorial evaluation, their contribution is not considered of enough interest or does not fall within the scope and politics of the journal.

However, we decided that it is reasonable to allow the authors an additional and last round of revision to answer the reviewer's comments. It is imperative that these language issues are resolved, otherwise we will be forced to reject the article.

With personal regards

Salvatore Andrea Mastrolia
PeerJ Academic Editor

Reviewer 4 ·

Basic reporting

The English in this manuscript is poor. Several grammar or spelling errors are noted. The followings are part of them:
Line 68:typing error “reported” instead “reproted”
Line 78: typing error “beginning” instead “begnning”
Line 80, 81: typing error “menstrual” instead “menstral”
Line179; typing error “cryopreservation” instead “cytopreservation”.

Experimental design

Line 63-67:The sentence and concept repeated in this section. Please correct it. The new added reference is not able to declare the “increased clinical pregnancy rate”. Please modify your description about the benefit of GnRH agonist long protocol and further define the purpose of this study to compare the full dose of GnRH agonist injection at follicular phase with that at the mid-luteal phase.

Validity of the findings

No comments.

Additional comments

No comments.

---

## Round 0.4 · accepted · Accept

Dear Authors,

I would like to compliment with you for the efforts provided in addressing the Reviewer's comments.

The manuscript has been considered suitable for publication and can be accepted in its current form.

With personal regards,

Salvatore Andrea Mastrolia

#